# Potentiometric E-Tongue System for Geosmin/Isoborneol Presence Monitoring in Drinkable Water

**DOI:** 10.3390/s20030821

**Published:** 2020-02-04

**Authors:** Larisa Lvova, Igor Jahatspanian, Luiz H.C. Mattoso, Daniel S. Correa, Ekaterina Oleneva, Andrey Legin, Corrado Di Natale, Roberto Paolesse

**Affiliations:** 1Department of Chemical Sciences and Technologies, University “Tor Vergata”, 00133 Rome, Italy; roberto.paolesse@uniroma2.it; 2Laboratory of Artificial Sensory Systems, ITMO University, 197101 St. Petersburg, Russia; drjie@mail.ru (I.J.); ekaterina.oleneva@inbox.ru (E.O.); a.legin@spbu.ru (A.L.); dinatale@uniroma2.it (C.D.N.); 3Nanotechnology National Laboratory for Agriculture (LNNA), Embrapa Instrumentation, Sao Carlos 13560-970, Brazildaniel.correa@embrapa.br (D.S.C.); 4Institute of Chemistry, St. Petersburg State University, 199034 St. Petersburg, Russia; 5Department of Electronic Engineering, University “Tor Vergata”, 00133 Rome, Italy

**Keywords:** taste-and-odor-causing compounds, geosmin, 2-methyl-isoborneol, potentiometric E-tongue, water potability assessment

## Abstract

A potentiometric E-tongue system based on low-selective polymeric membrane and chalcogenide-glass electrodes is employed to monitor the taste-and-odor-causing pollutants, geosmin (GE) and 2-methyl-isoborneol (MIB), in drinkable water. The developed approach may permit a low-cost monitoring of these compounds in concentrations near the odor threshold concentrations (OTCs) of 20 ng/L. The experiments demonstrate the success of the E-tongue in combination with partial least squares (PLS) regression technique for the GE/MIB concentration prediction, showing also the possibility to discriminate tap water samples containing these compounds at two concentration levels: the same OTC order from 20 to 100 ng/L and at higher concentrations from 0.25 to 10 mg/L by means of PLS-discriminant analysis (DA) method. Based on the results, developed multisensory system can be considered a promising easy-to-handle tool for express evaluation of GE/MIB species and to provide a timely detection of alarm situations in case of extreme pollution before the drinkable water is delivered to end users.

## 1. Introduction

Water quality control through continuous monitoring is an important analytical challenge, since water is one of the most essential natural resources for humanity [1]. Several directives and legislations governing drinkable water quality parameters are issued worldwide [2,3,4,5]. In these documents the main attention is given to the water characterization in terms of polluting species, which may provide serious risks to the health of consumers; however, an important issue is represented by the water organoleptic properties, which can in general give the water “potability”. It is evident that consumers would prefer obtaining a transparent and odorless water from their water supply, with no visible suspended species, and be able to consume tap water without any further pre-treatment like filtering, boiling, degassing, etc. Unfortunately, the quality of water provided for human use around the world is not always satisfactory; in many regions, and even entire countries, the degree of water pollution is extremely high, while in other places, the presence of some polluting compounds in water, even if not toxic, results in very unpleasant organoleptic characteristics, limiting its potability [6]. Although they are semi-volatile compounds, geosmin (trans-1,10-dimethyl-trans-9 decalol, GE) and isoborneol/2-methyl-isoborneol (IB/MIB) are found among the most taste and odor-causing pollutants of drinking water obtained from surface water (Figure 1) [7]. These compounds have a very strong, musty earthy taste and odor and arise in surface water as a result of several filamentous and cyanobacteria strains metabolites’ release and further degradation [8]; moreover GE/MIB have an extremely low odor threshold concentration (OTC) around 10–20 ng/L, while some people can notice their presence at levels up to 4 ng/L [9].

At present, there are no regulations on the maximum permitted amounts of GE/MIB in drinkable water, since these compounds are not associated to any negative effect on human health [7]. Nevertheless, due to the very unpleasant taste and odor of drinking water contaminated with GE/MIB, consumers may prefer to purchase bottled water instead of consuming water from public water supplies, thus increasing the amount of waste plastics and overall environment contamination [10]. Nowadays more than one million plastic bottles are bought every minute around the world, with an estimated 300 million tons of plastic every year [11]. The resistance of GE/MIB to elimination through conventional water treatment processes, such as home boiling and filtration, or by coagulation, sedimentation, and chlorination methods employed at water treatment stations, presents a challenging issue to find an appropriate analytical procedure to monitor and predict the presence of these pollutants in drinkable water at levels lower than the OTC level of GE/MIB, to protect both the water consumers and water suppliers.

The conventional instrumental methods, in particular gas chromatography–mass spectrometry (GC-MS), coupled with different extraction/enrichment techniques are widely employed nowadays for the analysis of GE/MIB [12,13,14]. Unfortunately, despite the evident benefits such as high selectivity and high analytical precision, GC-MS requires costly equipment, qualified personnel, it is time consuming and, especially in a case of GE/MIB detection having very low OTC levels, requires a sample enrichment and pre-concentration. Hence, the development of novel analytical techniques for express and non-costly GE/MIB detection is an important challenge. Recently, some alternative analytical methods were reported, owing to lower costs and the possibility of rapid GE/MIB detection in potable water; for instance, colorimetry with Tortelli–Jaffe bromine-based reaction [15], enzyme-linked immunosorbent assay (ELISA) for geosmin [16] and indirect competitive immunoassay for MIB [17,18], application of chemical sensors [19] and multisensor arrays, e-nose [20,21], and e-tongue [22] in particular. In the last systems, the chemical sensors with different transduction principles are employed in a combination with multivariate data analysis for GE/MIB analytes identification at levels close to the OTC, thus representing a relevant alternative to costly instrumental techniques for water quality assessment. In fact, in the last decade the application of chemical sensors and a multisensor system for rapid water quality evaluation is becoming more and more popular, since they may enable a low-cost and fast control of water safety, potability, and effectiveness of the purification process [23,24,25,26,27].

We previously demonstrated the application of a potentiometric E-tongue system for water toxicity estimation in terms of cyano-bacterial microcystin toxins (MCs) detection [28]. The importance of sensors selection and choice of an appropriate mathematical model relating to the response of the multisensor system and MCs content detected by the standard ultra-high-performance liquid chromatography method with diode array detection (UHPLC-DAD )and by the colorimetric enzymatic approach were shown, and the prediction of MCs content in drinkable waters at concentrations lower than the provisional guideline value of World Health Organization (WHO) of 1 mg/L was demonstrated [29]. In the present study we report the results of potentiometric E-tongue application for classification of drinkable water spiked with different concentrations of geosmin (GE) and 2-methyl-isoborneol (MIB) in order to monitor the presence of these compounds at OTC level and to provide a timely detection of alarm situations in case of extreme pollution before drinking water is delivered to the end user.

## 2. Materials and Methods

### 2.1. Reagents

Isoborneol (IB) was purchased from Sigma-Aldrich (São Paulo, Brazil), (±)-geosmin and 2-methylisoborneol (GE/MIB) (100 μg/mL in methanol) were purchased from Sigma-Aldrich (Rome, Italy). Dimethyl sulfoxide (DMSO), methanol (MeOH), and tetrahydrofuran (THF) solvents were obtained from Carlo Erba Reagents (Rome, Italy). Membrane components, high molecular weight Poly(vinyl chloride) (PVC), bis(2-ethylhexyl) sebacate (DEHS) plasticizer, tridodecylmethyl ammonium chloride (TDMACl), potassium tetrakis-(4-chlorophenyl)borate (TpClPBK) lipophilic additives, and nonactine ionophore were purchased from Sigma-Aldrich (Rome, Italy). THF was distilled prior to use and 5,10,15,20-tetraphenylporphyrin manganese(III) chloride ionophore (Mn(TPP)Cl) was synthesized and fully characterized according to the literature procedure [30]. Millipore grade water was used for aqueous solution preparation. All the other chemicals were of analytical grade and used without further purification.

### 2.2. Sensor System

Potentiometric multisensory system was composed of 8 sensors with two different types of sensing membranes: PVC-based solvent polymeric membranes doped with Mn(TPP)Cl (sensor A1) and nonactin (sensor C1) ionophores; and chalcogenide glass membranes doped with different metal salts (G2-Cu, G7-Tl, G8-Ag, G10-Cd, G11-Pb). The PVC-based solvent polymeric membranes were formed according to the previously reported method [31,32]. For this, all the membrane components (PVC 30–33 wt%, plasticizer 60–66 wt%, ion-exchanger 0.1–10 wt%, and 1 wt% of ionophore) were dissolved in THF. The membrane cocktails were then cast in a 24 mm i.d. glass ring on a glass slide and the solvent was evaporated overnight. Discs of 9 mm in diameter were then cut out from the parent membrane and fixed with 10 wt% of PVC in cyclohexanone glue onto hollow PVC tubes that served as electrode bodies. The tubes were filled with a 0.01 mol/L mixture of NaCl and salt solution containing the respective primary ions for every sensor: Cl^−^ for A1, and NH_4_^+^ for C1. The Ag/AgCl reference electrodes were immersed in the sensors’ inner solution to close the electrical circuit. Chalcogenide glass sensors were purchased from Sensor Systems (St. Petersburg, Russia). For commercial chalcogenide glass sensors, Cu-wire/Ag-paste solid contact was employed instead of the inner filling solution and the inner reference electrode. The potentials of sensors were measured versus a saturated calomel reference electrode (SCE, AMEL, Milan, Italy), in a standard two-electrode configuration cell. Potentiometric measurements were performed with LiquiLab (ECOSENS srl, Rome, Italy) high-impedance analog-to-digital potentiometer. Prior to measurements, the sensors were soaked in 0.01 mol/L NaCl aqueous solution for at least 24 h.

The response of the multisensory system to isoborneol was tested in 1.7 × 10^−10^−1.9 × 10^−6^ mol/L aqueous calibration solutions, prepared by the addition of calculated amounts of 1 μg/mL and 10 μg/mL stock solutions of IB in DMSO to Millipore grade water. Similarly, the response of the multisensory system to geosmin/2-methylisoborneol was tested in 1.5 × 10^−10^−1.2 ×10^−6^ mol/L aqueous calibration solutions, prepared by the addition of calculated amounts of 1 μg/mL and 10 μg/mL stock solutions of GE/MIB in MeOH to Millipore grade water. The schematic presentation of the experimental set-up and e-Tongue image is given in Figure 2. 

### 2.3. Water Samples

Tap water samples with added geosmin/2-methylisoborneol pollutants in six different concentrations corresponding to 20, 100, 250, 500, 1000, and 10,000 ng/L were investigated. Each concentration was tested five times, and fresh tap water without pollutants was measured as a reference sample. Tap water was from regular water supply of “Tor Vergata” zone of Rome, Italy. The samples were prepared in 125 mL plastic bottles with a screw cap directly prior to the first measurement and were stored in the refrigerator prior to subsequent measurements. In total, 35 water samples were analyzed; three measurement sessions were performed during which the samples were measured in a random order.

### 2.4. Data Processing 

Non-supervised principal component analysis (PCA) was used for data dispersion evaluation and samples identification [23].

Partial least squares discriminant analysis (PLS-DA) was applied for classification of tap water samples tainted with GE/MIB at two concentration levels: the same OTC order from 20 to 100 ng/L and at higher concentrations from 0.25 to 10 mg/L. The PLS regression method was applied to correlate the E-tongue response to known GE/MIB content in tested samples. The constructed PLS-DA and PLS models were validated using the one-leave-out cross-validation for first approximation; where the data set was representative enough and the random split test set was employed. 

The root mean square error of calibration (RMSEC), root mean square error of validation (RMSEV), and the correlation coefficient of predicted versus measured correlation line, R^2^, were used to evaluate the efficiency of the obtained PLS models. The chemometric treatment was performed with Unscrambler software (v. 9.7, 2007, CAMO Software AS, Oslo, Norway).

## 3. Results and Discussion

On the first step of the study the responses of E-tongue sensors were registered in the model solutions of IB and GE/MIB prepared on distilled water background in the concentration range from 25 ng/L to 300 μg/L in order to simulate the tap water with unpleasant organoleptic characteristics, and to evaluate the ability of the multisensor system to detect these pollutants at OTC cut-off level. For this purpose, we employed the E-tongue system reported in our previous research dedicated to the microcystin toxins screening in potable water [28]. Due to the frequent co-occurrence of microcystins and GE and MIB taste-and-odor compounds in drinking water [33], the E-tongue system may provide a simultaneous and important evaluation of water toxicity and organoleptic potability, thus signaling possible alarm situations corresponding to a high level of drinking water pollution prior to its delivery to end users.

The responses of several selected sensors from E-tongue array, namely polymeric membrane sensor C1, chalcogenide glass sensors G7-Tl, G8-Ag, and polycrystalline sensor A7 toward growing concentrations of GE/MIB are shown in Figure 3. The distinct and correlated variation of sensor potentials both for polymeric cation-sensitive sensor C1 and polycrystalline anionic membrane A7 should be noticed; such a variation is possibly a result of GE/MIB partitioning from analyzed aqueous media and further accumulation on the sensor membrane surface. In the case of chalcogenide glass electrodes, the significant potential drop occurs as a result of electrostatic interactions among GE/MIB (probably partially protonated) and positively (G8-Ag) or negatively (G7-Tl) charged membranes. Hence, through the combination of selected sensors responses and application of a suitable chemometric modeling method the possibility to track the taste-and-odor-causing compounds presence and content can be considered. 

With an aim to distinguish the presence of taste-and-odor-causing compounds in aqueous solutions we first evaluate the dispersion of sensor array data obtained in all tested samples with the PCA method. As seen in Figure 4, the exploratory PCA technique indicates the ability of the E-tongue system to clearly distinguish among aqueous solutions of MIB and GE (prepared from standard solution of 100 μg/mL in methanol) and solutions of pure methanol (MeOH) in water. At the same time, a greater influence of DMSO solvent (used to prepare IB stock solution) on E-tongue sensors’ response is obtained, and this causes some difficulties in a proper discrimination between solutions of IB in DMSO/water and aqueous solutions of DMSO. The highest loadings, and hence the highest influence on solutions discrimination, is found for anion-sensitive sensors A1, A7, nonactin-based cationic sensor C1, and Tl-doped chalcogenide-glass electrode G7 with distinct redox sensitivity.

Due to the sensors drift observed between different days of measurements, mathematical drift correction is required [29,34]; we normalize the sensors’ response considering the sensors readings in distilled water prior to the calibration as reference values. First, the differences between sensors readings in water during the first measurement day and the sensors’ readings in water background solution during the following measurements, D_H2O_, were calculated. Then the E-tongue sensors responses in IB and GE/MIB solutions are normalized as:S_sn_IB_ = R_IB_n_ − R_DMSO_n_ + D_H2O_(1)
S_sn_MIB_GE_ = R_MIB_GE_n_ − R_MeOH_n_ + D_H2O_(2)
where S_sn_IB_ and S_sn_MIB_GE_ are valued as normalized responses of sensor s in calibration solution n of known concentration of analyte; R_IB_n_ and R_MIB_GE_n_ are responses of the sensors in solutions of IB and MIB_GE respectively; R_DMSO_n_ and R_MeOH_n_ are responses in aqueous solutions of solvents, DMSO and MeOH correspondingly; D_H2O_ is sensors drift correction value. 

As seen from Figure 5, a clear separation of IB and GE/MIB solutions from pure MeOH and DMSO solvents is obtained for normalized data, especially along the PC1 axis; while the concentration gradient of odor-causing compounds is clearly noted along PC2. 

Moreover, the application of the partial squares linear regression method, PLS1, demonstrated the correlation between E-tongue sensor responses and IB and GE/MIB content in calibration solutions in a semilogarithmic scale (Figure 6). One-leave-out cross validation method is used due to the small size of the initial data set. The adequate predictive power of E-tongue was found for IB in DMSO solutions with R^2^_cal_ = 0.998 and R^2^_val_ = 0.906 (99.3% and 87.1% of total explained variance at calibration and validation steps, respectively; 5PCs). GE/MIB methanol/aqueous solutions have a lower validation correlation coefficient, R^2^_val_ = 0.695 (R^2^_cal_ = 0.998, total explained variance: 96.4% and 69.7% for calibration and validation respectively; 4PCs).

At the next step of our study, tests in tap water samples spiked with different concentrations of GE/MIB are performed in order for the PLS-DA model to discriminate tap water samples containing these compounds at two concentration levels: the same OTC order from 20 to 100 ng/L (class 1) and at higher concentrations from 0.25 to 10 mg/L (class 2). The PLS-DA was conducted on a set composed of 30 samples: 15 samples of class 1 and 15 samples of class 2, respectively. The confusion matrix of one-leave-out PLS-DA cross validation is reported in Table 1, while the three-dimensional (3D) score-plot of discrimination results is shown in Figure 7. In the table, rows indicate the expected GE/MIB spiked tap water class and columns correspond to those predicted.

Only one sample with high taste-and-odor-causing compounds concentration is misclassified, while another three samples could not be attributed to any class. The correct classification of 83% of samples is obtained and this preliminary result is satisfactory, considering the very low OTC level of the pollutants investigated.

## 4. Conclusions

A possibility to assess the taste-and-odor-causing pollutants, geosmin (GE) and 2-methyl-isoborneol (MIB) in drinkable water by means of potentiometric E-tongue was investigated. The preliminary tests have demonstrated the E-tongue utility to discriminate tap water contaminated with GE/MIB at low (20 to 100 ng/L) and high (0.25 to 10 mg/L) concentration levels. Obtained results permit to consider the developed E-tongue system as a promising easy-to-handle tool for assessment of GE/MIB species, and to provide a timely detection of alarm situations in case of extreme pollution by these compounds before drinking water is delivered to the end user in time to allow decision making which is of key importance in water treatment stations. Such a system, being installed in the potable water withdrawal and distribution systems, and connected to the control center through wireless networks, could be implemented for economical and real-time water quality monitoring.

## Figures and Tables

**Figure 1 sensors-20-00821-f001:**
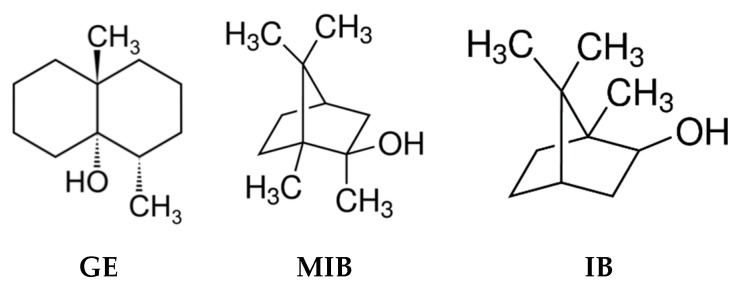
Chemical structures of taste-and-odor compounds: GE: geosmin, MIB: methyl-isoborneol, IB: isoborneol.

**Figure 2 sensors-20-00821-f002:**
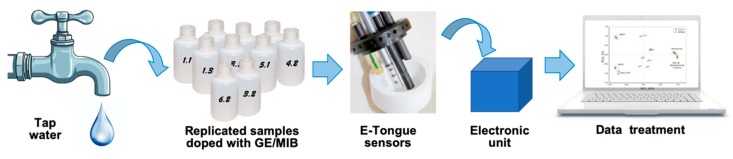
Schematic presentation of the employed measurement system.

**Figure 3 sensors-20-00821-f003:**
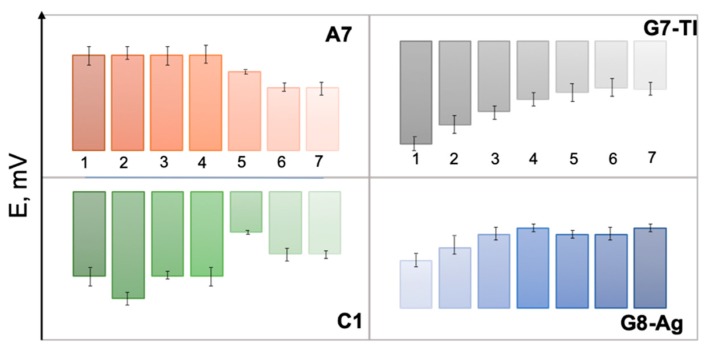
The responses of individual sensors A7, C1, G7-Tl, and G8-Ag in aqueous solutions of geosmin (GE) and 2-methyl-isoborneol (MIB) in concentrations (from left to right) (1): 25, (2): 50, (3): 100, (4): 500 ng/L, (5): 1, (6): 10, and (7): 100 μg/L.

**Figure 4 sensors-20-00821-f004:**
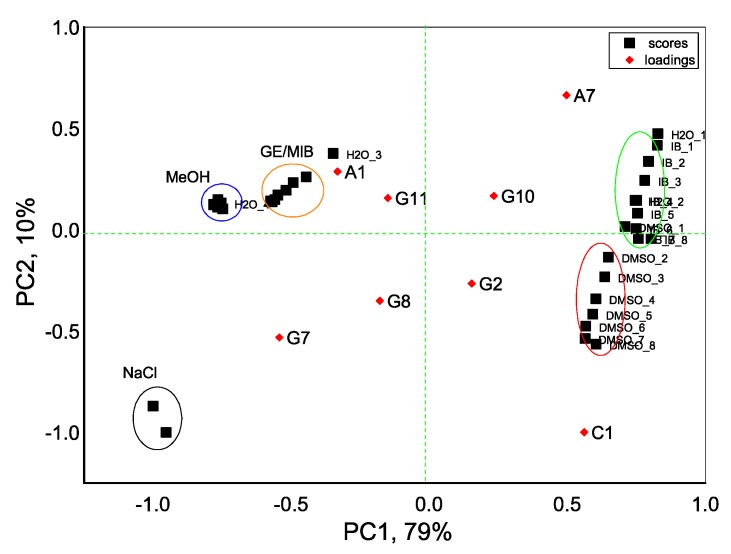
Principal component analysis (PCA) bi-plot result of E-tongue application for recognition of IB, DMSO, GE/MIB, and MeOH aqueous solutions in concentration range from 25 ng/L to 300 μg/L. The numbers in the sample labels correspond to the progressive measurement number.

**Figure 5 sensors-20-00821-f005:**
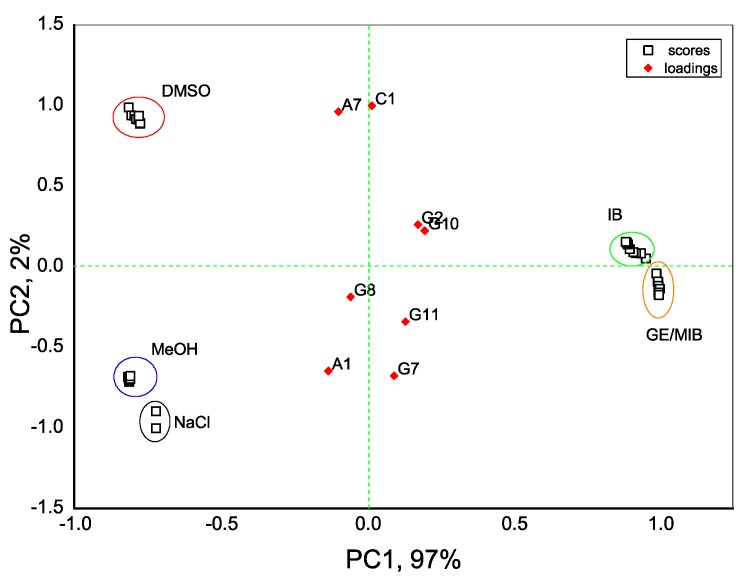
PCA bi-plot of E-tongue response in IB, DMSO, GE/MIB, and MeOH aqueous solutions after data normalization and solvent influence minimization.

**Figure 6 sensors-20-00821-f006:**
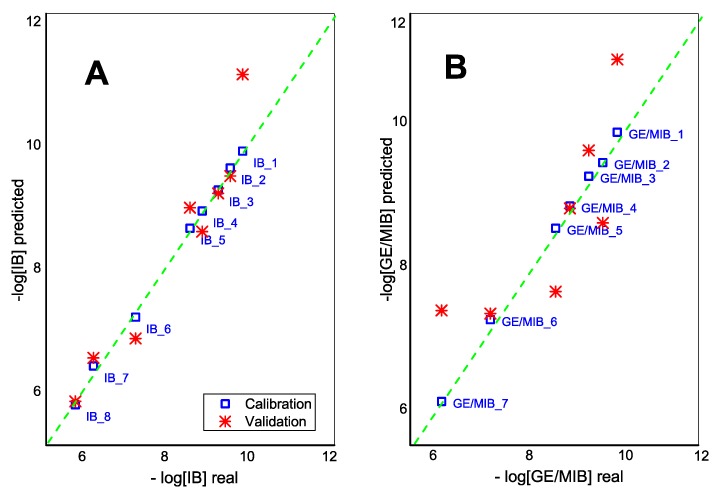
The PLS1 correlation result for (**A**) IB and (**B**) GE/MIB content determined by means of potentiometric E-tongue system.

**Figure 7 sensors-20-00821-f007:**
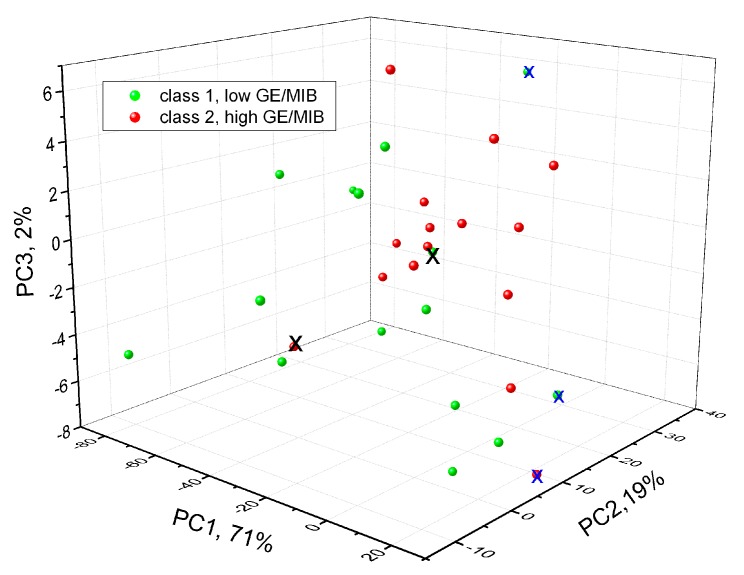
Three-dimensional (3D)-scores plot of PLS-DA GE/MIB detection in tap water by means of potentiometric E-tongue. X: misclassified samples; x: sample with no class attribution.

**Table 1 sensors-20-00821-t001:** Partial least square discriminant analysis (PLS-DA) confusion matrix of tap water samples classification to spiked GE/MIB concentration content.

Expected	Predicted	
	Class 1, Low Content (20–100 ng/ L)	Class 2, High Content (0.25–10 mg/L)
Class 1	13	1
Class 2	1	12
Non classified	2	1

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
