# Peer review of "Potentiometric E-Tongue System for Geosmin/Isoborneol Presence Monitoring in Drinkable Water"

_sensors, 2020, doi:10.3390/s20030821_

Round 1

Reviewer 1 Report

I have added the comments to the PDF file.

General suggestions include:

Introduction:

Elaborate on the importance of the detection of the substances described in comparision with other techniques. 

Elaborate on the descritpion of PCA

Methods:

Schematic or photo of the proposed methodology is required

Results:

Elaborate on the description of the phenomena. 

Conclusions

Elaborate on the impact of the results and future work. The methodology of how could be implemented

The overall quality of the work is good, but I think extensive editing and correction of the document are required. 

Please attend the comments on the PDF file

Author Response

General suggestions include:

Introduction:

Elaborate on the importance of the detection of the substances described in comparision with other techniques.

Actually the importance of new techniques development for GE/MIB detection is discussed in lines 69-84 of introduction. Nevertheless, in order to highlight the  importance of the detection analytes under discussion  in comparison with other techniques the new phrase was inserted in the manuscript text:

Hence, the development of the novel analytical techniques for express and non-costly GE/MIB detection it is an important challenge.

Elaborate on the description of PCA

PCA is a well known chemometric technique used for preliminary conclusion on data dispersion. There is no need to describe this method in introduction section, but some more details, as far as the relative reference [ref 23] were inserted in Data processing section of Material and Methods.:

"Non-supervised Principal Component Analysis (PCA) was used for data dispersion evaluation and samples identification [23]"

Methods:

Schematic or photo of the proposed methodology is required

The schematic representation of the system was inserted in the manuscript as Fig. 2.

Results:

Elaborate on the description of the phenomena. 

More details on possible sensors response mechanism, as far as in general description on phenomena were added in results section:

"The responses of several selected sensors from E-tongue array, namely polymeric membrane sensor C1, chalcogenide glass sensors G7-Tl, G8-Ag and polycrystalline sensor A7 toward growing concentrations of GE/MIB are shown in Figure 3. It should be noticed the distinct and correlated variation of sensor potentials both for polymeric cation-sensitive sensor C1 and polycrystalline anionic membrane A7; such a variation is possibly a result of GE/MIB partitioning from analyzed aqueous media and further accumulation on the sensor membrane surface. In the case of chalcogenide glass electrodes, the significant potential drop occurs as a result of electrostatic interactions among GE/MIB (probably partially protonated) and positively (G8-Ag) or negatively (G7-Tl) charged membranes. Hence, through the combination of selected sensors responses and application of a suitable chemometric modeling method the possibility to track the taste-and-odor-causing compounds presence and content can be considered"

Conclusions

Elaborate on the impact of the results and future work. The methodology of how could be implemented

Thanks for your valuable suggestion! We have inserted a Conclusions section in the manuscript, where the impact of the obtained results, as far as future works and possibilities of practical implementation od developed methodology are discussed in more details.

All the changes done  were yellow highlighted.

Reviewer 2 Report

I have reviewed the manuscript (#696125) entitled “Potentiometric E-tongue system for Geosmin/Isoborneol presence monitoring in drinkable water” submitted to Sensors. In this article, the authors demonstrate a potentiometry-based sensor combined with a multivariate data analysis technique and its application for the monitoring of chemical pollutants (=geosmin and 2-methyl-isoborneol) in drinkable water. Although the described approach for the simultaneous detection of target species is interesting for the development of on-site chemical sensing systems, I have several concerns with this research from the viewpoint of molecular recognition chemistry and analytical chemistry. Thus, I would like to recommend the acceptance of this article only after major revisions.  

1) In this article, the leave-one-out cross-validation protocol was performed to determine the level of correct classification of the observations within the PLS clusters (Section 2.4). However, the cross-validation result for the analysis of tap water samples was relatively low (= 83%), meaning that the cross-reactivity of the prepared sensor is not enough. The authors described that “…this preliminary result is satisfactory…”, but the origin of the obtained low-discrimination ability in the prepared sensor system should be discussed.

2) The authors should describe the design strategy of sensor materials for preparing multianalyte sensing systems utilizing multivariate data analysis techniques.

3) The authors should investigate the affinity between the utilized ionophores and the target analytes.

Author Response

1) In this article, the leave-one-out cross-validation protocol was performed to determine the level of correct classification of the observations within the PLS clusters (Section 2.4). However, the cross-validation result for the analysis of tap water samples was relatively low (= 83%), meaning that the cross-reactivity of the prepared sensor is not enough. The authors described that “…this preliminary result is satisfactory…”, but the origin of the obtained low-discrimination ability in the prepared sensor system should be discussed.

Thanks to referee for this comments, however, we should point out here, that the total system variance explained  by first 3 PCs during tap water samples spiked with GE/MIB was about 92% (see Figure 7 of the manuscript). During the classification only 2 samples over 30 (less than 7 % of total amount) were misclassified, and another 3 were not attributed to any of two classes. These data have permitted us to evaluate the percentage of proper classification 100%-((2+3)/30*100)% =  83.3%, and we we bielive that this result can be improved; and it was satisfactory for the actual step of our research and considering the complexity of the analytical task. The investigations on E-tongue sensor array composition that may further improve the final discrimination results on taste-and-odor-causing compounds assessment in potable water are now in progress in our laboratories.

2) The authors should describe the design strategy of sensor materials for preparing multianalyte sensing systems utilizing multivariate data analysis techniques.

The decision of E-tongue sensor array composition is given in lines 169-174 of the manuscript. Additionally, we have added some insights on the possible origin of E-tongue sensitivity to the tested taste-and-odor-causing compounds in lines 175-185 and in Fug.3.

3) The authors should investigate the affinity between the utilized ionophores and the target analytes.

As above-mentioned (see point 2), the interpretation of possible E-tongue sensors affinity to the tested taste-and-odor-causing compounds is given in lines 175-185 of the manuscript and in Fug.3.:

"The responses of several selected sensors from E-tongue array, namely polymeric membrane sensor C1, chalcogenide glass sensors G7-Tl, G8-Ag and polycrystalline sensor A7 toward growing concentrations of GE/MIB are shown in Figure 3. It should be noticed the distinct and correlated variation of sensor potentials both for polymeric cation-sensitive sensor C1 and polycrystalline anionic membrane A7; such a variation is possibly a result of GE/MIB partitioning from analyzed aqueous media and further accumulation on the sensor membrane surface. In the case of chalcogenide glass electrodes, the significant potential drop occurs as a result of electrostatic interactions among GE/MIB (probably partially protonated) and positively (G8-Ag) or negatively (G7-Tl) charged membranes. Hence, through the combination of selected sensors responses and application of a suitable chemometric modeling method the possibility to track the taste-and-odor-causing compounds presence and content can be considered"

The changes done  were yellow highlighted.

Reviewer 3 Report

The article "Potentiometric E-tongue system for Geosmin/ Isoborneol presence monitoring in drinkable water", by Lvova et al, describes the use of Potentiometric E-tongue system for Geosmin and Isoborneol detection in water samples. This work is interesting but a major revision of the manuscript needs to be done and a few points need to be further discussed.

1) In addition to the GE + MIB sample, why were samples with GE and MIB not analyzed separately?

2) If “people can notice their presence at levels up to 4 ng/L ”, why were the tests conducted with dilution only up to 20 ng/L and not below this value?

3) Section “2.4 Data Processing” did not mention the use of PCA.

4) After the normalization process, almost all of the information is described by the PC1 component, explaining why the point clouds associated with each solute became less scattered. Regarding the projection on the PC1 axis, the clouds associated with IB, NaCl, and MeOH did not change their position. However, MDSO and MIB_GE switched sides on the axis. There is now a clear separation of IB and MIB_GE solutions from DMSO solvent and now MIB_GE has clustered close to IB. Why does the proposed normalization not have the same effect on MDSO and MIB_Ge as it does on MeOH and IB?

5) Figure 4 does not appear to have been prepared for publication. It contains data and annotations that are not introduced and/or explained? What is blue and red data? What are the abscissa and ordinate axis data? What and why is the data on a semilog scale? This figure and its explanation need to be better elaborated.

6) Figure 5 does not illustrate the discrimination result. What is the aim of the plot?

Author Response

1) In addition to the GE + MIB sample, why were samples with GE and MIB not analyzed separately?

To perform our tests we have used the official standard solution from Sigma-Aldrich, that contains both (±)-geosmin and 2-methylisoborneol (GE/MIB) in amount of 100 μg/mL in methanol. The less soluble in water  isoborneol (IB), also purchased from Sigma-Aldrich , was tasted for comparison in order to evaluate the difference in E-tongue response towards single IB and GE/MIB mixture.  As it can be concluded from data reported in the manuscript, the developed potentiometric E-tongue is a promising easy-to-handle tool for assessment of both IB and  GE/MIB species.

2) If “people can notice their presence at levels up to 4 ng/L ”, why were the tests conducted with dilution only up to 20 ng/L and not below this value?

Actually, as it is mentioned in introduction, only some people may notice the presence of GE/MIB at extremely low level of 4 ng/L. These is not a case of overall population (and we may confirm it by our internal tests  performed on group of 10 people from the laboratory – among them no one has distinguished 4ng/L GE/MIB presence in tap water); hence , we have planned our experiments with an aim to asses  with E-tongue system the GE/MIB concentration at OTC level of  10-20 ng/L.    

3) Section “2.4 Data Processing” did not mention the use of PCA.

For correctness, the PCA method was mentioned in “2.4 Data Processing” section.

4) After the normalization process, almost all of the information is described by the PC1 component, explaining why the point clouds associated with each solute became less scattered. Regarding the projection on the PC1 axis, the clouds associated with IB, NaCl, and MeOH did not change their position. However, MDSO and MIB_GE switched sides on the axis. There is now a clear separation of IB and MIB_GE solutions from DMSO solvent and now MIB_GE has clustered close to IB. Why does the proposed normalization not have the same effect on MDSO and MIB_Ge as it does on MeOH and IB?

We believe, that the different solubility of IB and GE/MIG in two solvents, methanol and DMSO plays an important role also on the discrimination of  aqueous solutions of these compounds after normalization. In fact, since IB is poorly soluble in MeOH, the normalization permits obtain a clear separation of  last one and both solvents, while the similarity in IB and GE/MIB nature are now well seen through their close clustering.

5) Figure 4 does not appear to have been prepared for publication. It contains data and annotations that are not introduced and/or explained? What is blue and red data? What are the abscissa and ordinate axis data? What and why is the data on a semilog scale? This figure and its explanation need to be better elaborated.

Fig. 4 (in a new manuscript version it is Fig.6) was completely redrawn in order to better represent obtained PLS1 regression results on IB and GE/MIB concentrations prediction by means of E-tongue system. Moreover, the extended figure explanation was inserted in the manuscript text.

"The adequate predictive power of E-tongue was found for IB in DMSO solutions with R2cal=0.998 and R2val=0.906 (99.3% and 87.1% of total explained variance at calibration and vlidation steps respectively; 5PCs). In GE/MIB methanol/aqueous solutions a lower validarion correlation coefficient , R2val= 0.695 (R2cal = 0.998, total explained variance: 96.4% and 69.7% for   calibration and vlidation respectively; 4PCs)."  

6) Figure 5 does not illustrate the discrimination result. What is the aim of the plot?

Actually Figure 5 (the Figure 7 in new manuscript version) represents the 3D interpretation PLS-DA score plot result; where, actually even with a naked eye (and now also with X marks) the erroneously classified samples and samples non attributed to any class are well distinguished. The figure is in a good correlation with the main results of PLS-DA classification in a form of confusion matrix is summarized in Table 2. We would like to believe, that the changes made on Figure 7 will be useful for paper readers and approved by the accepted by referees.

Round 2

Reviewer 1 Report

The suggestions have been followed by the authors. Good work. 

Reviewer 2 Report

I satisfied all revisions by the authors. I would like to recommend the acceptance of this article as the current form.

Reviewer 3 Report

The authors answered the proposed questions. I recommend publishing the article.